# Change of Hypermucoviscosity in the Development of Tigecycline Resistance in Hypervirulent *Klebsiella pneumoniae* Sequence Type 23 Strains

**DOI:** 10.3390/microorganisms8101562

**Published:** 2020-10-10

**Authors:** Suyeon Park, Haejeong Lee, Dongwoo Shin, Kwan Soo Ko

**Affiliations:** Department of Microbiology and Samsung Medical Center, Sungkyunkwan University School of Medicine, Suwon 16419, Korea; parks024@naver.com (S.P.); hjlee7834@skku.edu (H.L.); shind@skku.edu (D.S.)

**Keywords:** tigecycline resistance, hypermucoviscosity, hypervirulent *Klebsiella pneumoniae*, ST23

## Abstract

In this study, we developed tigecycline resistance in *Klebsiella pneumoniae* ST23 strains in vitro and investigated the change in virulence associated with hypermucoviscosity. In vitro-induced tigecycline-resistant (TGC-IR) *K. pneumoniae* mutants were obtained from three tigecycline-susceptible (TGC-S) strains, belonging to ST23 and serotype K1, by culturing in media with tigecycline in a stepwise manner. An antimicrobial susceptibility test, string test, mucoviscosity assay, and capsular polysaccharide (CPS) quantification were performed. Biofilm formation and serum resistance were evaluated, and survival rates of bacterial strains in fruit flies and macrophages were measured. Alterations of *rpsJ, ramR, soxR, acrR*, and *marR* genes were investigated and the expression levels of *ramA* and efflux pump genes were evaluated. The hypermucoviscosity phenotype was dramatically decreased in the TGC-IR mutants. Reduced CPS production in TGC-IR mutants was also identified. Increased resistance to most other antimicrobial agents was found in TGC-IR mutants. In addition, the TGC-IR mutants exhibited reduced biofilm formation, low serum resistance, and decreased survival rates within fruit flies and macrophages. Our study shows that development of tigecycline resistance in hypervirulent *K. pneumoniae* strains result in defects in virulence associated with hypermucoviscosity.

## 1. Introduction

*Klebsiella pneumoniae* is a Gram-negative bacillus that is responsible for both community-acquired and nosocomial infections including pneumonia, urinary tract infections, bacteremia, and liver abscesses [1]. Since the mid-1980s, hypervirulent *K. pneumoniae* (hvKP), generally associated with the hypermucoviscosity (HV) phenotype, has emerged as a clinically significant pathogen responsible for serious disseminated infections [2,3]. The HV phenotype of hvKP is typically due to the increased production of capsular polysaccharide (CPS) and the presence of specific virulence genes [4,5]. Notably, sequence type 23 (ST23) is the most commonly described among hypermucoviscous *K. pneumoniae* isolates and is strongly correlated with capsular serotype K1 and liver abscess [1,6,7].

An increase in carbapenem resistance in *K. pneumoniae* worldwide has limited the available treatment options for the bacterium. Tigecycline is a last-resort antibiotic reserved for treatment in carbapenem-resistant *K. pneumoniae* infection [8]. Thus, tigecycline resistance means that options for the treatment of carbapenem-resistant *K. pneumoniae* infections are virtually eliminated. Tigecycline inhibits protein translation by binding to the 30S subunit of the bacterial ribosome, which ultimately prevents bacterial growth [9]. However, reports of tigecycline resistance have increased year by year [10]. In *K. pneumoniae*, resistance to tigecycline is mainly attributed to overproduction of efflux pumps (AcrAB, OqxA, KpgAB) [11] and to mutations in efflux pump regulator genes (*ramA*, *soxR, marR, acrR*) [12]. Mutations in *ramR*, which encodes a transcriptional repressor belonging to the TetR family, are also responsible for tigecycline resistance by leading to the overexpression of *ramA*. Moreover, mutations in *rpsJ*, the gene that encodes the ribosomal S10 protein, could confer reduced susceptibility to tigecycline in *K. pneumoniae* [13].

In this study, we investigated the association between tigecycline resistance and HV in hvKP isolates belonging to ST23. We described the decreased HV in three in vitro tigecycline-induced-resistant (TGC-IR) mutants of serotype K1 *K. pneumoniae* ST23 strains. We compared the phenotype, virulence, and CPS production in tigecycline-susceptible (TGC-S) and TGC-IR isolates. In addition, we performed qRT-PCR and sequencing of the genes related to tigecycline resistance in *K. pneumoniae*.

## 2. Materials and Methods

### 2.1. Bacterial Strains and In Vitro Induction of Tigecycline-Resistant Mutants

Three TGC-S *K. pneumoniae* strains, SMC1204-109, SMC1207-200, and SMC1208-086, were used in this study. They were obtained from patient blood samples at Samsung Medical Center (Seoul, Korea). The strains belonged to ST23 in multilocus sequence typing analysis. For TGC-S strains, MLST and K serotyping were performed as described previously [14,15]. The TGC-S strains were experimentally evolved to TGC-IR mutants using a previously described method [16]. Briefly, the three TGC-S *K. pneumoniae* strains were grown overnight at 37 °C and subcultured in Luria–Bertani (LB) broth with a serially increasing concentration of tigecycline (0.5 to 64 mg/L). The successfully developed TGC-IR mutants, SMC1204-109-IR, SMC1207-200-IR, and SMC1208-086-IR, were used in this study.

### 2.2. Antimicrobial Susceptibility Testing

Antimicrobial susceptibility testing was performed using the broth microdilution method following Clinical and Laboratory Standards Institute (CLSI) guidelines [17]. The minimum inhibitory concentrations (MICs) of ten antimicrobial agents including tigecycline, tetracycline, ampicillin, gentamicin, cefotaxime, ceftazidime, meropenem, amikacin, ciprofloxacin, piperacillin-tazobactam, and colistin were determined. The MICs of tigecycline in the presence of efflux pump inhibitors, carbonyl cyanide 3-chlorophenylhydrazone (CCCP), and phenyl-arginine-β-naphthylamide (PAβN) were also measured. Antimicrobial susceptibility was defined according to the CLSI breakpoints [17], with *Escherichia coli* ATCC 25922 and *Pseudomonas aeruginosa* ATCC 27853 as reference strains. Tigecycline MICs were interpreted according to EUCAST guidelines [18]. Tigecycline susceptibility of bacteria was defined using only *Escherichia coli* ATCC 25922 as the reference strain. All tests were performed in duplicate with three biological replicates per strain.

### 2.3. String Test

The HV phenotype of the TGC-S and TGC-IR strains was evaluated using the string test as described previously [1]. All tested isolates were cultured overnight on blood agar plates at 37 °C, and bacterial colonies were then stretched with an inoculation loop.

### 2.4. Mucoviscosity Assay and CPS Quantification

Because the supernatant of HV strains remains turbid after centrifugation, measurement of the turbidity after centrifugation can serve as a quantitative indicator of HV [19]. Therefore, the mucoviscosity assay was performed following a previously described methodology [20]. Bacterial strains were grown in LB broth at 37 °C overnight with shaking. The samples were centrifuged at low speed (1000× *g*) for 5 min and the absorbance of supernatants was measured at 600 nm (OD_600_).

To quantify CPS in TGC-S and TGC-IR *K. pneumoniae* strains, we extracted and measured the amount of CPS from bacteria as described previously [16]. Bacterial cultures were mixed with 1% Zwittergent 3-14 detergent (Sigma-Aldrich, St. Louis, MO, USA) in 100 mM citric acid and then, the mixtures were incubated at 50 °C for 30 min. After centrifugation, supernatants were transferred into new tubes, and 1 mL of absolute ethanol was added. The pellets were dissolved in 100 μL of distilled water, and then, 600 μL of 12.5 mM boric acid in H_2_SO_4_ was added to each tube. The mixtures were vigorously vortexed, incubated at 100 °C for 5 min, and then, cooled. Next, 10 μL of 3-hydroxydiphenol (Sigma-Aldrich, St. Louis, MO, USA) was added to the mixture and the absorbance was measured at 520 nm. The glucuronic acid concentration in each sample was determined from a standard curve of glucuronic acid and expressed as micrograms per 10^9^ CFU/mL.

### 2.5. Biofilm Formation and Serum Resistance Assays

To evaluate biofilm formation, 96-well microtiter plate assays were performed with crystal violet as described previously [21], with minor modifications. Briefly, the mid-log phase bacterial cultures were added to a flat bottom plate and incubated at 37 °C. After 24 h, all of the bacterial cultures and LB broth (used as the negative control) were removed and washed twice with phosphate-buffered saline (PBS). Then, 0.1% crystal violet solution was added to the plate for staining before incubation for 20 min at room temperature. The solution was completely removed and washed three times with PBS. The walls were dried, and bound dye was solubilized with 200 μL of 95% ethanol and quantified by an xMark Microplate Absorbance Spectrophotometer (Bio-Rad, Hercules, CA, USA) at 540 nm. Each assay was performed in duplicate and repeated three times independently.

The serum resistance assay was performed as described previously [22]. Normal human serum diluted to 20% (NHS, Innovative Research, Novi, MI, USA) was treated to the mid-log phase bacterial cultures. As a control, heat-inactivated human serum (HIS) was used to determine the bactericidal effect of NHS. After 3 h of incubation with shaking, the mixtures were serially diluted and plated on blood agar. The number of colony-forming units (CFUs) that survived after treatment with NHS was compared with the number of CFUs that survived after treatment with HIS. All assays were performed three times and the results are represented as survival percentage.

### 2.6. Drosophila Melanogaster (Fruit Fly) Infection

To assess the virulence of TGC-S and TGC-IR strains, a fruit fly infection model was used with the thoracic needle-prick method as described previously [22]. Three- to five-day-old female flies were infected with bacterial cultures (OD_600_ of 20) using an ultra-fine needle (BD Biosciences, San Jose, CA, USA). A pure PBS injection was used as a negative control. Twenty flies were infected for each bacterial strain; the mortality of flies was monitored until 72 h post infection. The experiments were performed three times independently.

For quantification of viable bacteria, infected flies from each bacterial strain group were individually immersed in 100 μL of PBS. Five flies from each strain were homogenized by a Teflon pestle and serially diluted. Then, the mixtures were plated onto LB agar containing 100 mg/L of ampicillin. The number of CFUs per fly was counted.

### 2.7. Macrophage Infection Assay

A macrophage infection assay was performed to assess the survival rate of bacteria using a previously described method [23], with slight modifications. Macrophage-like line J774A.1 cells were grown in Dulbecco’s Modified Eagle Medium (DMEM) (Welgene, Gyeongsan, Korea) supplemented with 10% fetal bovine serum (FBS) (Gibco) and 1% antibiotic–antimycotic solution (Thermo, Waltham, MA, USA). A monolayer of macrophage cells was prepared in a 12-well tissue culture plate before bacterial infection. Macrophage cells (1 × 10^6^) were washed with Dulbecco’s phosphate-buffered saline (DPBS) (Welgene, Gyeongsan, Korea) and incubated in DMEM with FBS for 1 h. Bacterial isolates cultured overnight were inoculated at ratio of 20 bacteria per macrophage (MOI 20). The cells were incubated for 30 min at 37 °C to permit phagocytosis and the supernatants were washed with DPBS three times. After infection with bacterial strains, pre-warmed DMEM supplemented with a high concentration of gentamicin (150 mg/L) was added to kill extracellular bacteria and incubated for 1 h. The wells for the 0-h time point were washed with DPBS and treated with 1% Triton X-100 immediately. For the 4- and 20-h time point samples, the washed cells were supplemented with a low concentration of gentamicin (15 mg/L) and treated with Triton X-100 at each time point. Then, the diluted mixture from each well was spread on blood agar plates for the calculation of CFUs. The survival percentages of bacteria after 4 and 20 h were calculated compared to the CFUs at the 0 h time point. All experiments were performed in duplicate and repeated three times independently.

### 2.8. Sequencing for Assessing Gene Mutations

For analysis of nucleotide alterations in *ramR, marR, acrR, soxR,* and *rpsJ,* we sequenced the genes with primers described previously [12,24]. Genomic DNA was extracted from the *K. pneumoniae* strains using a gDNA extraction kit (iNtRON, Seongnam-si, Korea) according to the manufacturer’s instructions. Sequencing of the PCR products was conducted by Macrogen (Seoul, Korea).

### 2.9. Quantitative RT-PCR

To compare the alteration of gene expression according to tigecycline resistance in *K. pneumoniae*, quantitative real-time PCR (qRT-PCR) was performed [25]. Total RNA was extracted from mid-log phase bacterial cultures (OD_600_ approximately 0.5) using the Qiagen RNeasy Mini kit (Qiagen, Hilden, Germany) according to the manufacturer’s instructions. After removal of contaminating DNA from the RNA samples, reverse transcription reactions were performed using the reverse transcription premix kit (iNtRON, Seongnam-si, Korea). To quantify the target genes, qRT-PCR was performed using TB Green Premix Ex Taq (TaKaRa, Shiga, Japan) on the QuantStudio 6 Flex Real-Time PCR system (Applied Biosystems, CA, USA) with the primers listed in Appendix A. The fold-changes were calculated according to the comparative threshold cycle (ΔΔC_T_) method, using the *rpoB* gene as the reference. The experiments were repeated with three independent cultures and each sample was tested in duplicate.

### 2.10. Statistical Analyses

Statistical analyses were performed using Prism version 3.00 for Windows (GraphPad Software, San Diego, CA, USA). The differences were assessed using Student’s *t*-test, one-way ANOVA with Tukey’s multiple comparisons test, and a nonparametric Kruskal–Wallis test followed by Dunnett’s multiple comparison test. A *p*-value of less than 0.05 was considered statistically significant (*, *p* < 0.05; **, *p* < 0.001; ***, *p* < 0.0001).

## 3. Results

### 3.1. Decreased Hypermucoviscosity in TGC-IR Mutants

All three TGC-S *K. pneumoniae* strains used in this study, SMC1204-109, SMC1207-200, and SMC1208-086, were identified as serotype K1. TGC-IR mutants were developed from TGC-S strains by culturing in media including tigecycline in a stepwise manner, showing tigecycline MICs of >64 mg/L (Table 1).

The HV phenotype was observed in the three TGC-S *K. pneumoniae* strains in the string test, ranging from 25 to 35 mm (Figure 1). The length of the bacterial string was reduced remarkably in the TGC-IR mutants, to 8 mm in SMC1204-109-IR, 5 mm in SMC1207-200-IR, and 4 mm in SMC1208-086-IR (Figure 1). Such lengths were 11.4% to 32.0% of those in their parental strains.

The mucoviscosity was also investigated by measuring the absorbance of supernatants after low-speed centrifugation (Figure 2A). Consistent with the results of the string test, all three TGC-IR mutants exhibited significantly reduced turbidity, indicating low mucoviscosity, compared with that of their parental TGC-S *K. pneumoniae* strains. In addition, the quantity of CPS was also significantly lower (approximately 10- to 54-fold) in the TGC-IR mutants than in the TGC-S strains (Figure 2B).

### 3.2. Antimicrobial Susceptibility Profiles

The three TGC-S *K. pneumoniae* strains were susceptible to most antimicrobial agents except ampicillin (Table 2). All TGC-IR mutants showed increased MICs for tetracycline, ampicillin, cefotaxime, and piperacillin-tazobactam. While the MICs of ceftazidime, meropenem, ciprofloxacin, and colistin also increased in two TGC-IR mutants, SMC1204-109-IR and SMC1207-200-IR, it did not in the other strain, SMC1208-086-IR. The MIC of amikacin in SMC1208-086-IR was reduced compared with that of its parental strain, and gentamicin MICs were not changed in all TGC-IR mutants.

The MICs for tigecycline with efflux pump inhibitors, CCCP and PAβN, were also measured. For the parental TGC-S *K. pneumoniae* strains, the tigecycline MIC was reduced only in SMC1204-109 with CCCP and in SMC1208-086 with PAβN (Table 2). Their induced-resistance mutants maintained very high tigecycline MICs, 64 mg/L or higher, when 50 μM of CCCP or 25 mg/mL of PAβN was added to the media.

### 3.3. Biofilm Formation and Serum Resistance

Biofilm formation decreased in three TGC-IR strains compared with that of their parental TGC-S *K. pneumoniae* strains, although the difference was significant only in SMC1204-109-IR (*p* = 0.0425) (Figure 3A).

The survival rates of the TGC-S strains and TGC-IR mutants were evaluated in the presence of NHS over a 3-h period. The survival rates of the three TGC-IR mutants were lower than those of their parental TGC-S strains (Figure 3B). Although the reduction in survival was not significant in one mutant (SMC1204-109-IR), it was significant in the other two (*p =* 0.0131 and 0.0462).

### 3.4. Fruit Fly and Macrophage Infection

Survival rates of *D. melanogaster* (fruit fly) against TGC-S and TGC-IR *K. pneumoniae* strains were evaluated (Figure 4A–C). While SMC1204-109-IR showed significantly decreased fly-killing ability compared with that of its parent strain (*p* < 0.005) (Figure 4A), the other TGC-IR mutants did not (Figure 4B,C). In addition, we evaluated the viable bacteria inside the fruit flies after 72 h of infection. The infected flies were homogenized and plated to calculate the CFU of infecting bacteria (Figure 4D). While many viable colonies (1 × 10^3^ to 5 × 10^5^ CFU/fly) were identified in TGC-S strains, no viable bacteria were found when the TGC-IR mutants were administered.

The survival of the bacterial strains inside macrophages was also evaluated (Figure 5). The survival rates of three TGC-IR mutants were significantly lower than those of their parental TGC-S strains after 20 h of macrophage infection.

### 3.5. Alterations in rpsJ, ramR, soxR, acrR, and marR Genes in TGC-IR Mutants

Amino acid variations in the *rpsJ* gene (ribosomal S10 protein) and transcriptional regulator genes (*ramR, soxR, acrR*, and *marR*) in TGC-IR mutants were investigated. Amino acid alteration in RpsJ was identified only in SMC1207-200-IR (Table 3) [26]. Mutations in the *ramR* gene were identified in all three TGC-IR mutants, but the mutations were different between them. These three mutations in the *ramR* gene cause a frameshift and premature stop: a deletion of four nucleotides at position 8 to 11 in SMC1204-109-IR, an insertion of adenine at position 30 in SMC1207-200-IR, and a deletion of eight nucleotides at position 128 to 135 in SMC1208-086-IR (Table 3). SMC1207-200-IR had an additional mutation in the *soxR* gene, an insertion of cytosine at position 94. No alterations were identified in *acrR* or *marR* genes between TGC-S and TGC-IR strains.

### 3.6. Expression of Genes related to Tigecycline Resistance

We measured the mRNA expression levels of genes that are known to be associated with tigecycline resistance in *K. pneumoniae*, using the qRT-PCR method. The transcriptional activator, RamA, was up-regulated significantly in all three TGC-IR mutants, compared with that of their parental TGC-S strains (*p*-values of 0.0059, 0.0009, and 0.0014) (Figure 6A). However, the expression levels of the efflux pump genes, *acrB*, *oqxB,* and *kpgB*, were not different between TGC-S strains and TGC-IR mutants (Figure 6B).

## 4. Discussion

In the present study, we report that the acquisition of tigecycline resistance may lead to the decreased HV phenotype, resulting in decreased virulence in *K. pneumoniae* clinical strains. Three tigecycline-susceptible HV *K. pneumoniae* isolates used in this study belonged to ST23 and showed the K1 serotype. Because hypermucoviscous *K. pneumoniae* are highly invasive, they have emerged as clinically important pathogens responsible for serious infections [3]. Serotype K1 is the most common among the hypermucoviscous *K. pneumoniae* [1], and the close association with the serotype and genotype ST23 is well-known [7,27]. Thus, the emergence of antimicrobial resistance in this pathogenic clone should be a concern [28].

Reduced hypermucoviscosity in tigecycline resistance-induced *K. pneumoniae* ST23 strains was identified in both the string test and mucoviscosity assay. The decreased mucoviscosity was closely associated with reduced production of CPS. Although the HV phenotype and CPS production are well-known determinants of virulence in *K. pneumoniae* [4], decreased virulence in tigecycline resistance-induced ST23 strains was verified, with low serum resistance and low survival rates of bacteria within fruit flies and macrophages.

Several studies have revealed that antimicrobial resistance in HV *K. pneumoniae* strains is relatively low compared to that of non-HV strains [5,29]. The three clinical strains included in this study were also susceptible to most antimicrobial agents except ampicillin. However, the development of tigecycline resistance leads to increased resistance to other antimicrobial agents, although the increase was lower than for tigecycline. Thus, some universal changes in the development of tigecycline resistance may affect susceptibility to other antimicrobial agents.

In a previous study, it was reported that the acquisition of colistin resistance in HV *K. pneumoniae* ST23 strains was accompanied by reduced capsule production, impaired virulence, and a significant fitness cost [16]. However, no change in resistance to other antimicrobial agents was observed in that study. Thus, the mechanisms responsible for decreased virulence accompanied with reduced HV and CPS production may be different between the processes of induction of tigecycline and colistin resistance.

Many studies have suggested that the efflux pump AcrAB contributes to tigecycline resistance in *K. pneumoniae* [12,30]. It was also shown that AcrAB can be up-regulated by *ramR* mutations and subsequently by *ramA* activation [31]. We identified mutations in *ramR* and increased transcription of the *ramA* gene in tigecycline resistance-induced mutants. However, no significant increase in the expression of *acrAB* or other efflux pump genes was identified, and efflux pump inhibitors such as CCCP and PAβN did not significantly decrease tigecycline MICs. Thus, efflux pumps may not contribute to tigecycline resistance and the increase in MICs for other antimicrobial agents in our *K. pneumoniae* mutants. Our results may indicate that *ramA* overexpression is necessarily associated with the function of efflux pumps, and that efflux pumps may not contribute to tigecycline resistance and the increase in MICs for other antimicrobial agents in our *K. pneumoniae* mutants. Elucidation of the key factors contributing to both antimicrobial resistance and virulence in *K. pneumoniae* is needed.

In summary, we found that the development of tigecycline resistance in hvKP strains may result in the decrease or loss of the HV phenotype and virulence. In addition, the induction of tigecycline resistance may be associated with an increase in resistance to other antimicrobial agents. Although development of tigecycline resistance may be due to RamA up-regulation associated with *ramR* mutations, efflux pumps may not be responsible for the phenomenon. Although our study has limitations that in vitro tigecycline-induced-resistant mutants were investigated and only three strains were included, the mechanisms of reduced virulence in the development of tigecycline resistance are potential targets to control hvKP infections. Thus, more investigation is required on the determinants of virulence associated with tigecycline resistance in *K. pneumoniae*.

## Figures and Tables

**Figure 1 microorganisms-08-01562-f001:**
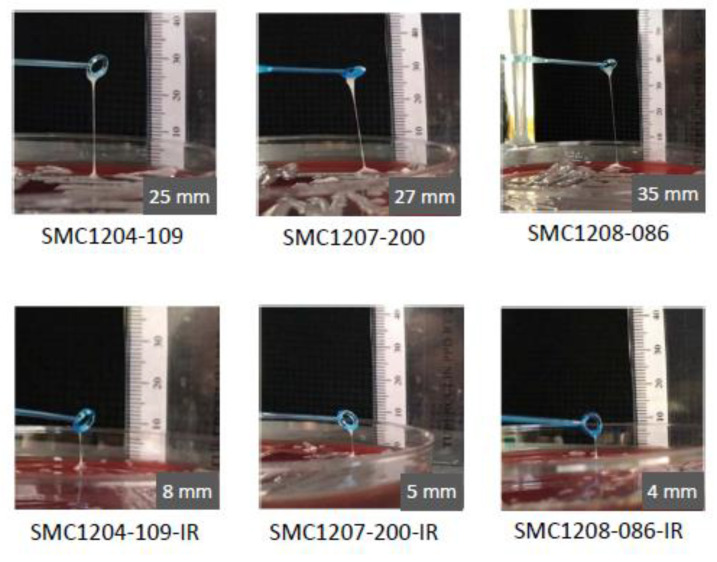
Results of string tests. Comparison of stretched colonies between TGC-S and TGC-IR *K. pneumoniae* strains. Generally, a string 5 mm or longer was defined as positive.

**Figure 2 microorganisms-08-01562-f002:**
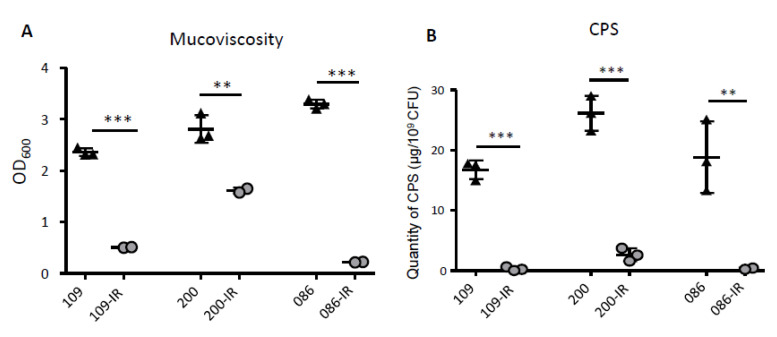
Results of mucoviscosity assay and capsular polysaccharide (CPS) production. (**A**) Mucoviscosity of TGC-S and TGC-IR isolates was measured. The absorbance of supernatants was compared after low-speed centrifugation. (**B**) Production of CPS was quantified in the TGC-S isolates and TGC-IR mutants. CPS biosynthesis was determined by phenol-sulfuric acid assays. **, *p* < 0.001; ***, *p* < 0.0001.

**Figure 3 microorganisms-08-01562-f003:**
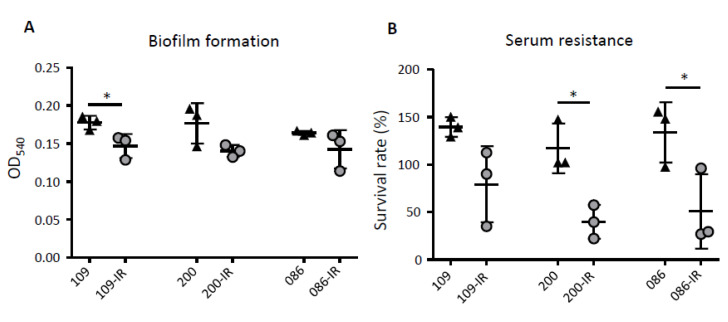
Results of biofilm formation and serum resistance assays. (**A**) Biofilm formation of TGC-S and TGC-IR *K. pneumoniae* strains. The biofilm was stained with crystal violet and quantified by measuring the absorbance at 540 nm. (**B**) The survival rate of *K. pneumoniae* strains was determined after 3 h of incubation with human serum. Heat-inactivated serum (HIS) was used as a negative control. *, *p* < 0.05.

**Figure 4 microorganisms-08-01562-f004:**
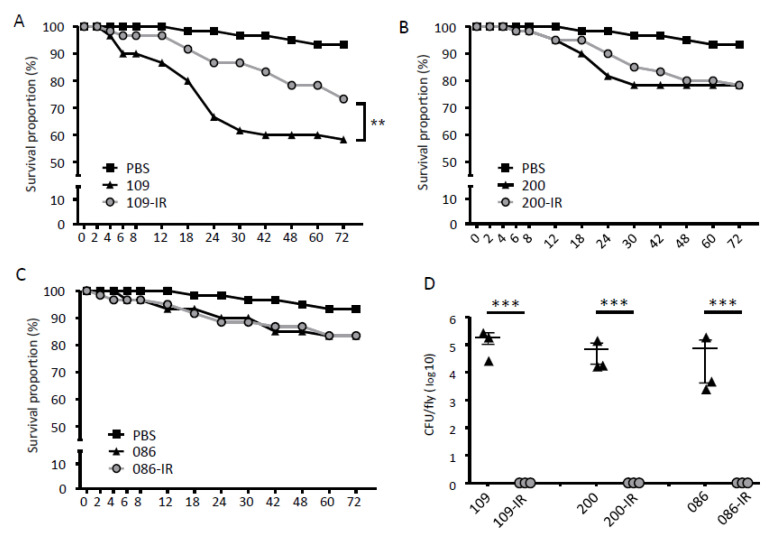
Results of fruit fly infections. (**A**–**C**) Survival of flies infected with TGC-S and TGC-IR strains. Fifteen flies were infected with each bacterial strain in mid-log phase (OD_600_ = 0.5). (**D**) The number of surviving colonies of bacterial strains in the flies after 72 h of infection. The dots indicate the number of CFUs in a single fly. ***, *p* < 0.0001.

**Figure 5 microorganisms-08-01562-f005:**
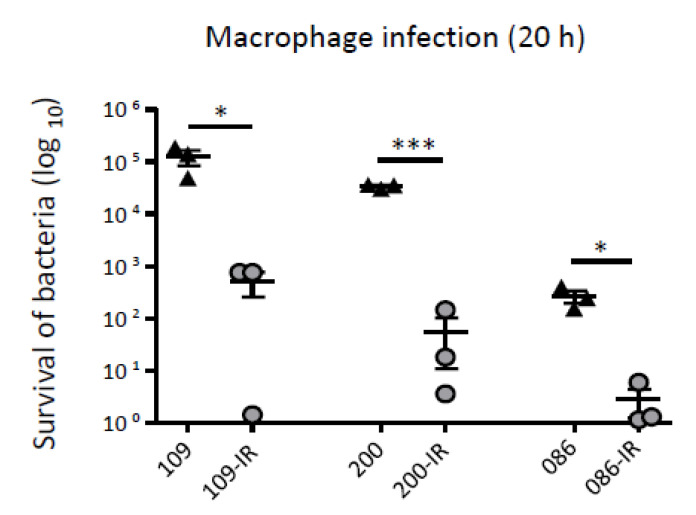
Result of macrophage infection assay. Survival rates of TGC-S and TGC-IR strains inside macrophages (J774A.1). The survival percentage of bacteria after 20 h was calculated by comparing to the CFUs at the 0 h time point. *, *p* < 0.05; ***, *p* < 0.0001.

**Figure 6 microorganisms-08-01562-f006:**
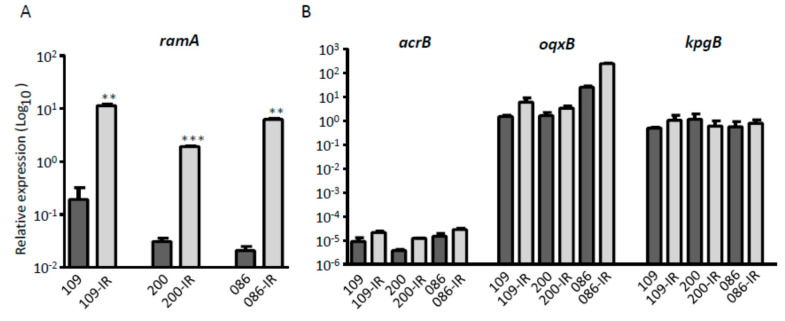
Expression levels of genes related to tigecycline resistance. (**A**) The expression level of the transcriptional activator gene *ramA*. (**B**) The expression levels of three efflux pump genes, *acrB*, *oqxB*, and *kpgB*. The fold-changes were calculated using the comparative threshold cycle (ΔΔC_T_) method. **, *p* < 0.001; ***, *p* < 0.0001.

**Table 1 microorganisms-08-01562-t001:** Tigecycline minimum inhibitory concentrations (MICs) and the results of the string test in *Klebsiella pneumoniae* strains used in this study.

Strains	Strain No.	Tigecycline MIC ^a^	String Test
TGC-S strains	SMC1204-109	1 (S)	25 mm
	SMC1207-200	0.5 (S)	27 mm
	SMC1208-086	2 (S)	35 mm
TGC-IR mutants	SMC1204-109-IR	>64 (R)	8 mm
	SMC1207-200-IR	>64 (R)	5 mm
	SMC1208-086-IR	>64 (R)	4 mm

^a^ R—resistant; S—susceptible.

**Table 2 microorganisms-08-01562-t002:** Antimicrobial susceptibility profiles of *Klebsiella pneumoniae* strains used in this study.

Strain	MIC (mg/L) (Antimicrobial Susceptibility Category) ^a, b^												
	TGC	TGC + CCCP ^c^	TGC + PaβN ^d^	TET	AMP	GEN	CTX	CAZ	MRP	AMK	CIP	P/T	CL
	TGC-S strains												
SMC1204-109	1 (S)	0.5 (S)	0.5 (S)	2 (S)	64 (R)	128 (R)	0.125 (S)	1 (S)	0.06 (S)	1 (S)	0.06 (S)	4/4 (S)	0.25 (S)
SMC1207-200	0.5 (S)	0.5 (S)	0.5 (S)	2 (S)	64 (R)	128 (R)	0.125 (S)	0.5 (S)	0.06 (S)	1 (S)	0.06 (S)	4/4 (S)	0.25 (S)
SMC1208-086	2 (S)	2 (S)	1 (S)	4 (S)	64 (R)	128 (R)	0.125 (S)	1 (S)	0.06 (S)	1 (S)	0.06 (S)	4/4 (S)	0.5 (S)
	TGC-IR mutants												
SMC1204-109-IR	>64 (R)	64 (R)	64 (R)	>64 (R)	>64 (R)	128 (R)	2 (I)	4 (S)	0.125 (S)	1 (S)	0.5 (S)	64/4 (I)	1 (S)
SMC1207-200-IR	>64 (R)	>64 (R)	>64 (R)	>64 (R)	>64 (R)	128 (R)	4 (R)	4 (S)	0.125 (S)	0.5 (S)	0.5 (S)	64/4 (I)	1 (S)
SMC1208-086-IR	>64 (R)	64 (R)	64 (R)	32 (R)	>64 (R)	128 (R)	1 (S)	1 (S)	0.06 (S)	0.25 (S)	0.06 (S)	8/4 (S)	0.5 (S)

^a^ TGC—tigecycline; TET—tetracycline; AMP—ampicillin; GEN—gentamicin; CTX—cefotaxime; CAZ—ceftazidime; MRP—meropenem; AMK—amikacin; CIP—ciprofloxacin; P/T—piperacillin-tazobactam; CL—colistin. ^b^ R—resistant; I—intermediate-resistant; S—susceptible. ^c^ CCCP—carbonyl cyanide 3-chlorophenylhydrazone. ^d^ PAβN—phenyl-arginine-β-naphthylamide.

**Table 3 microorganisms-08-01562-t003:** Mutations in the *rpsJ* gene and transcription regulator genes in TGC-IR mutants compared with that of their parental TGC-S strains.

Genes	SMC1204-109-IR	SMC1207-200-IR	SMC1208-086-IR
*rpsJ*	−	V57L	−
*ramR*	4-bp deletion at nt 8 to 11	A insertion at nt 30	8-bp deletion at nt 128 to 135
*soxR*	−	C insertion at nt 94	−
*acrR*	−	−	−
*marR*	−	−	−

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
