# Peer review of "Change of Hypermucoviscosity in the Development of Tigecycline Resistance in Hypervirulent Klebsiella pneumoniae Sequence Type 23 Strains"

_microorganisms, 2020, doi:10.3390/microorganisms8101562_

Round 1

Reviewer 1 Report

The manuscript "Change of hypermucoviscosity in the development of tigecycline resistance in hypervirulent Klebsiella pneumoniae sequence type 23 strains" is devoted to the study of phenotypic changes in Klebsiella strains accompanying the development of tigecycline resistance. In general, it is well and logically written and conclusions are confirmed by the results obtained. At the same time, there are several comments that require clarification from the authors of the manuscript.

Please add a brief description of the function of the investigated protein RamR, it is not described in the introduction.

Please explain why did you use gentamicin to kill extracellular bacteria (section 2.7), but gentamicin-resistance in tigecycline-resistant Klebsiella mutants was not checked. You showed that TGC-R Klebsiella strains become resistant to a number of antibiotics (section 3.2), please add the data concerning resistance to gentamicin.

Please add any explanation for the fact that high RamA expression does not lead to the activation of efflux pump genes.

Minor remarks:

Page 2, line 73 "accoording". Please, correct.

Page 6 line 205 contains a mistake in the name of the concentration of the CCCP (mM/μl), please correct the name of the concentration.

Author Response

  1. Please add a brief description of the function of the investigated protein RamR, it is not described in the introduction.

- As suggested, we described the function of RamR in the Introduction.

“Mutations in ramR, which encodes a transcriptional repressor belonging to the TetR family, are also responsible for tigecycline resistance by leading to the overexpression of ramA.” (Line 51-52 in the revised manuscript)

  1. Please explain why did you use gentamicin to kill extracellular bacteria (section 2.7), but gentamicin-resistance in tigecycline-resistant Klebsiella mutants was not checked. You showed that TGC-R Klebsiella strains become resistant to a number of antibiotics (section 3.2), please add the data concerning resistance to gentamicin.

- The gentamicin MICs of K. pneumoniae strains including TGC-S and TGC-IR were 128 mg/L. Thus, the concentration of gentamicin (150 mg/L) used to kill extracellular bacteria was enough. We included the data in the revised manuscript (including Table 2)

“The minimum inhibitory concentrations (MICs) of ten antimicrobial agents including tigecycline, tetracycline, ampicillin, gentamicin, cefotaxime, ceftazidime, meropenem, amikacin, ciprofloxacin, piperacillin-tazobactam, and colistin were determined.” (Line 77 in the revised manuscript)

“The MIC of amikacin in SMC1208-086-IR was reduced compared with that of its parental strain, and gentamicin MICs were not change in all TGC-IR mutants.” (Line 208-209 in the revised manuscript)

  1. Please add any explanation for the fact that high RamA expression does not lead to the activation of efflux pump genes.

- As suggested, we added explanation on the relationships between RamA overexpression and efflux pumps.

“Our results may indicate that ramA overexpression is necessarily associated with the function of efflux pumps, and that efflux pumps may not contribute to tigecycline resistance and the increase of MICs for other antimicrobial agents in our K. pneumoniae mutants.” (Line 288-291 in the revised manuscript)

Minor remarks:

  1. Page 2, line 73 "accoording". Please, correct.

- We corrected it.

  1. Page 6 line 205 contains a mistake in the name of the concentration of the CCCP (mM/μl), please correct the name of the concentration.

- we corrected it (50 μM of CCCP).

Reviewer 2 Report

 The authors stated clearly what study found and how they did it. The references are relevant and recent. Appropriate and key studies are included.

The study methods are valid and reliable. There are enough details provided in order to replicate the study. The study design is appropriate to answer the aim. This study added to what is already in the topic.

The data is presented in an appropriate way. The text in the results add to the data and it is not repetitive. Statistically significant results are clear. Results are discussed from different angles and placed into context without being overinterpreted.

The conclusions answer the aim of the study. The conclusions are supported by references and own results.

The article is consistent within itself.

Major

  1. The introduction can be extended with a paragraph regarding the clinical importance of tigecycline resistance

Minor

  1. What are the limitations of the study, if any?
  2. What is the clinical significance and usefulness of the obtained results?

Author Response

Major

  1. The introduction can be extended with a paragraph regarding the clinical importance of tigecycline resistance

- As suggested, we extended the paragraph on the clinical importance of tigecycline resistance.

“Increase of carbapenem resistance in K. pneumoniae worldwide has limited the available treatment options for the bacterium. Tigecycline is a last-resort antibiotic reserved for treatment in carbapenem-resistant K. pneumoniae infection [8]. Thus, tigecycline resistance means that options for treatment of carbapenem-resistant K. pneumoniae infections are virtually eliminated.” (Line 43-46 in the revised manuscript)

Minor

  1. What are the limitations of the study, if any?

- Our study investigated the tigecycline resistance mechanisms in in vitro tigecycline induced resistant isolates, not in clinical isolates. In addition, we investigated only three strains. We think that they are limitations of our study. We mentioned them in revised manuscript.

“Although our study has limitations that in vitro tigecycline-induced-resistant mutants were investigated and only three strains were included,” (Line 297-299 in the revised manuscript)

  1. What is the clinical significance and usefulness of the obtained results?

- As indicated in submitted manuscript, reduced virulence in the development of tigecycline resistance may be potential targets to control hvKP infections. Because tigecycline is a last resort for treatment of carbapenem-resistant K. pneumoniae infections, reduced virulence associated with acquired tigecycline resistance would lead to the development of anti-virulence drug. The former was mentioned in the original manuscript, and the latter is too overestimated, thus we did not add the sentence in the revised manuscript.

Round 2

Reviewer 1 Report

All corrections were done and I have no more comments. I think it may be accepted in the present form.